# MET-PREVENT: metformin to improve physical performance in older people with sarcopenia and physical prefrailty/frailty – protocol for a double-blind, randomised controlled proof-of-concept trial

Katherine J Rennie,[1] Miles Witham ![ORCID],[2] Penny Bradley,[3] Andrew Clegg ![ORCID],[4] Stephen Connolly,[5] Helen C Hancock ![ORCID],[1] Shaun Hiu,[6] Leanne Marsay,[1] Claire McDonald,[7] Laura Robertson,[1] Laura Simms,[1] Alison J Steel,[1] Claire J Steves,[8] Bryony Storey,[7] James Wason ![ORCID],[6] Nina Wilson ![ORCID],[6] Thomas von Zglinicki,[9] Avan A P Sayer[2]

KJR and MW are joint first authors.

For numbered affiliations see end of article.

**Correspondence to**
Professor Miles Witham;
miles.witham@newcastle.ac.uk

## ABSTRACT

**Introduction** Skeletal muscle dysfunction is central to both sarcopenia and physical frailty, which are associated with a wide range of adverse outcomes including falls and fractures, longer hospital stays, dependency and the need for care. Resistance training may prevent and treat sarcopenia and physical frailty, but not everyone can or wants to exercise. Finding alternatives is critical to alleviate the burden of adverse outcomes associated with sarcopenia and physical frailty. This trial will provide proof-of-concept evidence as to whether metformin can improve physical performance in older people with sarcopenia and physical prefrailty or frailty.

**Methods and analysis** MET-PREVENT is a parallel group, double-blind, placebo-controlled proof-of-concept trial. Trial participants can participate from their own homes, including completing informed consent and screening assessments. Eligible participants with low grip strength or prolonged sit-to-stand time together with slow walk speed will be randomised to either oral metformin hydrochloride 500 mg tablets or matched placebo, taken three times a day for 4 months. The recruitment target is 80 participants from two secondary care hospitals in Newcastle and Gateshead, UK. Local primary care practices will act as participant identification centres. Randomisation will be performed using a web-based minimisation system with a random element, balancing on sex and baseline walk speed. Participants will be followed up for 4 months post-randomisation, with outcomes collected at baseline and 4 months. The primary outcome measure is the four metre walk speed at the 4-month follow-up visit.

**Ethics and dissemination** The trial has been approved by the Liverpool NHS Research Ethics Committee (20/NW/0470), the Medicines and Healthcare Regulatory Authority (EudraCT 2020-004023-16) and the UK Health Research Authority (IRAS 275219). Results will be made available to participants, their families, patients with sarcopenia, the public, regional and national clinical teams, and the international scientific community.

## STRENGTHS AND LIMITATIONS OF THIS STUDY

⇒ Seeks to enrol older people with sarcopenia and physical prefrailty or frailty—an underserved group.
⇒ Wide-ranging mechanistic studies embedded in the trial.
⇒ Flexible recruitment methods and flexible study visits including option for participants to be seen at home.
⇒ Limitations of short-term follow-up and small sample size.
⇒ Will exclude some people with multimorbidity due to safety considerations.

**Trial registration number** ISRCTN29932357.

## BACKGROUND

Skeletal muscle dysfunction is central to both sarcopenia and physical frailty, which are common syndromes in older people. Sarcopenia, the loss of muscle strength and mass that commonly accompanies ageing[1] is a major health problem for many older people. Sarcopenia is a major risk factor for falls, hospitalisation, increased length of stay, care home admission and earlier death.[2–4] In addition, sarcopenia is an important component of the physical frailty syndrome, which is associated with a similar range of adverse outcomes.[5] Physical frailty is typically defined as the presence of three or more of five physical characteristics (weight loss, low energy expenditure, exhaustion, slow gait speed and low grip strength).[6] The presence of one or two of

these characteristics indicates the presence of physical prefrailty as a precursor to physical frailty. Impaired physical performance is therefore a key characteristic of both sarcopenia and physical frailty and is a target for intervention that is prioritised by patients.[7]

At present, resistance exercise training is the only intervention proven to improve outcomes for people with sarcopenia or physical frailty, with limited evidence on its effects in physical prefrailty.[8–10] Not all older people with sarcopenia or physical frailty are able or willing to undertake resistance training. As a result, alternative therapeutic options are needed to both prevent and improve sarcopenia and physical frailty, including through targeting physical prefrailty as a precursor state to more advanced physical frailty. Although the aetiology of sarcopenia and physical frailty are incompletely understood, it is becoming clear that multiple fundamental biological pathways related to ageing are important in driving these syndromes. Key pathways include inflammation, mitochondrial dysfunction, neuromuscular junction dysfunction, cellular senescence, and dysregulation of nutrient sensing and intracellular metabolism.[1 11]

Metformin, a biguanide molecule, has been used as a treatment for type 2 diabetes mellitus for decades; it is a generally safe and well-tolerated therapy even in older people with physical frailty or multiple health conditions. Importantly, the mode of action of metformin in lowering glucose in patients with type 2 diabetes does not rely on stimulating insulin release, and thus users are not subjected to an increased risk of hypoglycaemia. Metformin has pleiotropic effects on glucose metabolism and energy utilisation as well as on a range of other age-related pathophysiological pathways; several of these actions may be beneficial in treating the muscle dysfunction seen in sarcopenia and physical frailty as summarised in figure 1.[12–23] Not all mechanisms of action of metformin may be beneficial however. Some pathways (eg, adenosine monophosphate kinase activation) are similar to those triggered by caloric restriction,[24] which while potentially beneficial for longevity over a long period may have adverse catabolic consequences in the shorter term. It is only by testing metformin in older people with sarcopenia that the balance of benefits and risks can be properly assessed.

Observational studies show lower rates of cardiovascular events and cancer mortality in metformin users,[25 26] consistent with a wide range of physiological effects as outlined above. Few studies have examined the relationship between metformin use, sarcopenia and physical frailty. Such observational studies are challenging because metformin is currently indicated for use in diabetes mellitus—itself implicated in accelerated ageing, skeletal muscle dysfunction and earlier onset of physical frailty. However, incident and prevalent frailty-related diseases (including falls, weight loss, gait disorders and frailty diagnosed using a cumulative deficits Frailty Index, FI) have been found to be less common in patients with type 2 diabetes treated with metformin compared with those

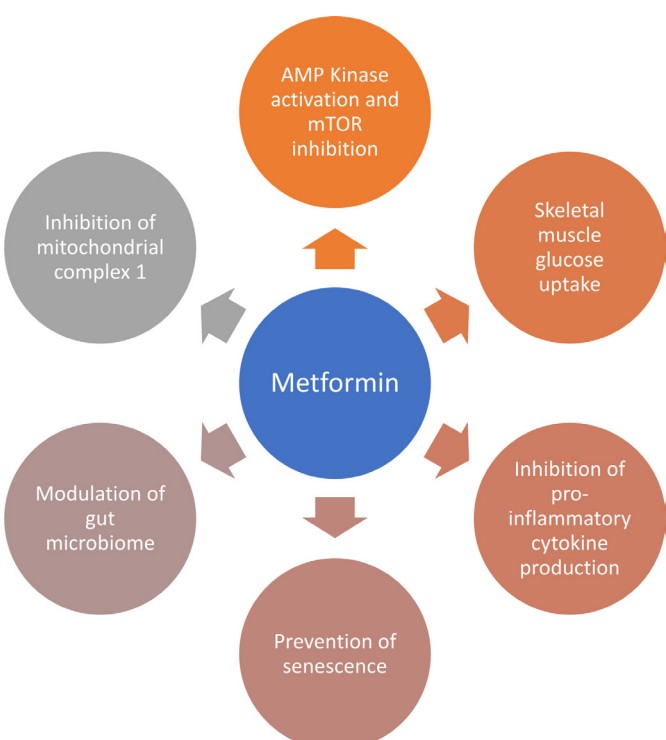

**Figure 1** Potential mechanisms of action through which metformin could improve skeletal muscle function. AMP Kinase, Adenosine MonoPhosphate activated protein Kinase; mTOR, mammalian Target Of Rapamycin.

treated with other glucose-lowering agents in observational studies.[26 27]

One previous trial conducted in Indonesia studied people aged 60 and over with prefrailty (defined by either the Fried criteria or by a cumulative deficits index) but without diabetes.[28] Participants were randomised to receive 16 weeks of 500 mg metformin thrice daily or matched placebo. Walk speed increased significantly in the treatment group compared with placebo (by 0.13 m/s; which exceeds the minimum clinically important difference of 0.10 m/s[29]). The recent MASTERS randomised trial did not show evidence that metformin enhanced the effect of resistance training, but the focus of this trial (augmentation of resistance training) was different and did not target people with sarcopenia or physical frailty.[30]

## Trial objectives

MET-PREVENT is a parallel group, randomised, double-blind, placebo-controlled trial. The primary objective of MET-PREVENT is to provide proof of concept evidence as to whether metformin is superior to placebo in improving physical performance in older people with sarcopenia and physical prefrailty or frailty. The secondary objectives are to elucidate potential mechanisms of action of metformin on sarcopenia and physical frailty by correlating changes in biomarkers with changes in physical performance measures between baseline and 4 months. A placebo-controlled design has been selected to minimise bias and is appropriate given the absence of other pharmacological interventions for sarcopenia or physical frailty.

## METHODS: PARTICIPANTS, INTERVENTIONS AND OUTCOMES

### Trial setting

Participants will be recruited from the catchment areas of two secondary care hospitals in the North-East of England. Participants will be approached through Older Peoples Medicine Clinics, Day Units, assessment units and rehabilitation facilities at the hospitals. In addition, local general practices (GPs) will act as participant identification centres, a registry of patients with sarcopenia[31] and the National Institute for Health and care Research BioResource will be used to identify additional potential participants. A flexible approach to the setting for participation will be offered, with the option of either research centre, clinic or the participant's own home as the setting for trial visits.

### Eligibility criteria

The target population for MET-PREVENT is older adults (≥65 years) with sarcopenia and either physical prefrailty or frailty. Sarcopenia is defined using the handgrip strength and sit to stand thresholds from the 2019 European Working Group on Sarcopenia in Older People (EWGSOP) guidance.[32] Low muscle mass is not used as an eligibility criterion in this trial; the 2019 EWGSOP guidance allows a diagnosis of probable sarcopenia without muscle mass measurement. Muscle mass is not routinely measured in clinical practice, and the omission of this criterion will improve the relevance of the trial results to clinical practice. Physical prefrailty and frailty are defined following the method of Fried et al[6]; participants will have a minimum of two of the five Fried criteria (and thus fulfil the definition for prefrailty) but may have three or more criteria (fulfilling the definition of frailty). The MET-PREVENT eligibility criteria are listed in box 1. Exclusion criteria are primarily designed to exclude participants at higher risk of adverse events (AEs) from taking metformin, those where metformin therapy is already indicated (eg, diabetes mellitus) and other types of skeletal myopathy (eg, steroid myopathy, heart failure myopathy) where confusion with sarcopenia might arise.

### Interventions

All randomised participants will take either metformin hydrochloride (500 mg film-coated) tablets or matched placebo tablets, orally, three times a day with food or just after a meal. No dose adjustments are planned. Study medication is dispensed in a single bottle which is identical for metformin and placebo.

Investigators may discontinue the trial treatment in the event of side effects occurring, that are possibly, probably or definitely related to trial medication and which are not tolerable to the participant, or which constitute a serious adverse reaction (SAR) or suspected unexpected SAR. Treatment will also be discontinued if the participant requests the medication to be withdrawn; in the event of a new diagnosis of diabetes mellitus (type 1 or type 2) or symptomatic chronic heart failure; if the estimated glomerular filtration rate (GFR) falls below 30 mL/min/1.73 m²

---

**Box 1   MET-PREVENT inclusion and exclusion criteria**

**Inclusion criteria**
⇒ Adults aged ≥65 years.
⇒ Low maximum handgrip strength (<16 kg for women, <27 kg for men) OR 5× sit to stand time >15 s.
⇒ Slow walk speed (<0.8 m/s on 4 me walk test).

**Exclusion criteria**
⇒ Diabetes mellitus (type 1 or type 2).
⇒ Previous intolerance of metformin or taking metformin for another condition.
⇒ Any contraindication to metformin, as listed in the current summary of medicinal product characteristics for metformin.
⇒ Any medication which significantly interacts with metformin, as listed in the current summary of medicinal product characteristics for metformin.
⇒ Estimated glomerular filtration rate <45 mL/min/1.73 m² by the Modified Diet in Renal Disease 4 (MDRD4) or Chronic Kidney Disease - EPIdemiology collaboration (CKD-EPI) equation.
⇒ History of diarrhoeal illness within the last three months (>48 hours of Bristol stool chart grade 6 or 7).
⇒ Alcohol intake >21 units/week (women) or >35 units/week (men).
⇒ Symptomatic chronic heart failure, diagnosed according to European Society of Cardiology guidelines (asymptomatic left ventricular systolic dysfunction will not be an exclusion criterion).
⇒ Liver function tests (bilirubin, alanine aminotransferase or alkaline phosphatase) >3× upper limit of normal.
⇒ Oral steroid dose >7.5 mg prednisolone equivalent per day.
⇒ Unable to mobilise without human assistance.
⇒ Unable to give written informed consent.
⇒ Life expectancy of <3 months as adjudicated by the local Investigator.
⇒ Participation in other interventional studies within 30 days prior to trial entry. Coenrolment with other interventional studies is not allowed (observational studies and registries are permitted.

---

or plasma lactate concentration is >4 mmol/L in the safety follow-up visit blood tests, or if lactic acidosis is diagnosed as part of clinical workup for an intercurrent illness. The trial medication may be temporarily discontinued at the instigation of the treating clinical team or the study team if an episode of acute kidney injury or an intercurrent illness with a risk of dehydration or hypovolaemia occurs. Once this has resolved, trial medication may be restarted if the estimated GFR is >30 mL/min/1.73 m². Participants who discontinue allocated trial medication and wish to remain in the trial will be followed up as per their allocated treatment intervention arm. If a participant does not want to attend any further trial visits but is willing to complete questionnaires by phone/post then they may continue in the trial.

### Outcomes

#### Primary outcome

The primary outcome is the between-group difference in the four metre walk speed at 4 months. The four metre walk speed is performed from a standing start, with the participant asked to walk at their usual pace. Short-course walking speed is a powerful predictor of a range of adverse outcomes in older people, including death,

dependency and cognitive impairment.[33–35] The 4 m walk speed has been used as an outcome measure in previous trials, including those of metformin[30] and the minimum clinically important difference for this measure has been estimated to be 0.1 m/s.[28]

### Secondary outcome measures

Secondary outcome measures in the MET-PREVENT trial include other measures of physical performance, health-related quality of life, mechanistic outcomes and outcomes related to trial performance. Maximum hand-grip strength will be measured using a Jamar dynamom-eter[36]; three readings will be taken for each hand and the highest reading used. A 6 min walk distance will be recorded over a 10 m course with standardised encouragement.[37] The Short Physical Performance Battery (SPPB)[38] will be measured as a composite test of lower limb function with the five-times sit to stand time performed as part of the SPPB analysed as an additional, separate outcome. Lean body mass will be estimated using the Akern 101 bioimpedance system (Akern, Pontassieve, Italy), with the Sergi equation used to derive measures of total lean body mass.[39] The number of physical frailty characteristics (0–5) and the components of the score, and transitions between robust (0 characteristics), physical prefrailty (1–2 characteristics) and physical frailty (3+ characteristics) will be recorded.

Generic health-related quality of life will be recorded using the EuroQoL 5 dimension 5 level (EQ5D-5L) and Short-Form 36 (SF-36) (Physical and Mental Component Summary Scores) questionnaires.[40 41] Instrumental activities of daily living (ADLs) will be recorded using the Nottingham extended activities of daily living (NEADL) score.[42] Glycosylated haemoglobin will be measured along with the homeostatic measure of insulin resistance (HOMA-IR) derived from peripheral glucose and insulin measures.[43] Advanced glycosylation end-products (AGE) presence in the skin will be measured by auto fluorescence using the AGE Reader (Diagnostics, Gronigen, Netherlands).[44] Blood samples will be processed and stored for later mechanistic studies, including measurement of a panel of proinflammatory cytokines and markers of cellular senescence and oxidative stress. Stool samples will be collected and stored for later microbiome studies. Metrics describing the conversion rate from screening to randomisation, and the retention rate of recruited participants over their 4-month study participation will also be collected.

### Participant timeline

Participants who express interest in the trial will enter a brief pre-screening process by telephone to check provisional eligibility. Verbal consent will be sought for the prescreen process, including access to medical records for further review of suitability for the trial. At prescreening, participants are asked if they have a diagnosis of diabetes mellitus, are taking metformin, or have previously been intolerant of metformin. The Strength, Assistance, Rise, Climb, Falls (SARC-F) questionnaire will be administered[45]; this comprises five questions on physical function with a score between 0 and 10. Based on previous data,[46] a score of 1 or more will be sufficient to identify those more likely to have sarcopenia and will enable progression to the screening visit. Participants who pass the prescreen will be given or posted the full participant information sheet (PIS), and all participants will be given at least 48 hours to consider their participation. The PIS can be found in online supplemental appendix 1.

Written informed consent will be obtained at the screening visit, which may take place at home, at clinic or in a research facility. After informed consent is given, information on demographics, medical and medication history and alcohol use will be collected. Maximum hand-grip and 4 m walk speed will be measured. Blood will be taken for urea, creatinine and electrolytes (U&Es) and liver function tests (LFTs) unless results are already available from within 3 months prior to the screening visit. Participant eligibility to proceed to the baseline visit is confirmed after the screening visit, once screening assessment results are available.

The schedule of events is given in table 1. Main study outcomes are measured at baseline and the 4-month final visit; safety visits to check renal function, lactate concentrations and AEs take place at 1, 2 and 3 months, and a telephone call takes place 1 week after the baseline visit to ensure that the participant has received and started the study medication.

### Sample size

Based on previous work, we assumed a minimum clinically important difference for the four metre walk speed of 0.1 m/s.[28] Assuming a SD of 0.24 as seen in the English Longitudinal Study of Ageing,[47] and a correlation between baseline and 6-month measures of 0.8, as seen in a recent trial of older people at risk of falls[48] (a similar population to those who will be enrolled in this trial), a total of 66 participants would be needed (33 per group) to give 80% power to detect this difference at a two-sided alpha=0.05 in an analysis that is adjusted for baseline values. Recruiting 80 participants will allow for 17.5% drop-out (higher than observed in previous trials of spironolactone and of vitamin D in similar populations[49 50]) to give 66 participants at follow-up.

### Recruitment

Recruitment will take place over ten months and participants will be followed up for 4 months; recruitment started in September 2021. Potential participants for MET-PREVENT will be identified by four routes. The SarcNet Registry[31] contains details of participants who have had grip strength and walk speed measured previously and have consented to be recontacted with offers to participate in other studies. The NIHR BioResource Centre Newcastle registry contains details of participants who have previously given consent for recontact with information about potential trial opportunities suitable

**Table 1** MET-PREVENT trial schedule of events

| Timepoint | Prescreening | Screen | Baseline | 1 week (+3days) after randomisation | 1 month (+1 week) after randomisation | 2 months (+1 week) after randomisation | 3 months (+1 week) after randomisation | 4 months ±2 weeks |
|---|---|---|---|---|---|---|---|---|
| Screening, consent and preallocation assessments | | | | | | | | |
| Brief study information sheet and invitation letter posted/given to patients identified via SarcNet Registry, older peoples medicine clinics, day units and rehabilitation facilities and GP practices | X | | | | | | | |
| Telephone prescreening (verbal consent, demographics, SARC-F tool) for participants who send a positive reply slip | X | | | | | | | |
| Patient information sheet posted to participants who pass prescreen | X | | | | | | | |
| Informed written consent | | X | | | | | | |
| Eligibility assessments (U&Es, LFTs, demographics, medical history, concomitant medication, adverse events, four metre walk speed, 5 x sit to stand time, grip strength) | | X | | | | | | |
| Eligibility confirmation | | X | | | | | | |
| Baseline assessments preallocation (blood glucose and HbA1c, height and weight, bioimpedance, AGE skin fluorescence, physical performance tests (SPPB, grip strength, six min walk), quality of life questionnaires (EQ5D5L, SF-36, Nottingham EADL), frailty screening questions (activity, exhaustion). | | | X | | | | | |
| Allocation (randomisation)—following completion of assessments | | | X | | | | | |
| Dispensing/posting trial medication | | | X | | | | | |
| Biological samples | | | | | | | | |
| Blood and stool | | | X | | | | | X |
| Intervention | | | | | | | | |
| 4 months supply of Metformin hydrochloride 500 mg tablets or matching placebo tablets | | | ← | | | | → | |
| Follow-up | | | | | | | | |
| Confirm medication receipt, ability to open medication bottle, date of first dose | | | | X | | | | |
| Safety blood tests (U&Es, LFTs, glucose, lactate) | | | | | X | X | X | X |
| Adverse event assessments | | X | X | X | X | X | X | X |
| Concomitant Medications | | X | X | X | X | X | X | X |
| Follow-up assessments (blood glucose and HbA1c, weight, bioimpedance, AGE skin fluorescence, physical performance tests (SPPB, grip strength, 6 min walk), quality of life questionnaires (EQ5D5L, SF-36, Nottingham EADL), frailty screening questions (activity, exhaustion). | | | | | | | | X |
| Return of unused trial medication | | | | | | | | X |
| Medication compliance, accountability | | | | | | | | X |

AGE, advanced glycosylation end; EADL, extended activities of daily living; EQ5D5L, EuroQoL 5 dimension 5 level; GP, general practices; HbA1c, glycated haemoglobin; LFTs, liver function tests; SARC-F, Strength, Assistance, Rise, Climb, Falls; SF-36, short form 36; SPPB, Short Physical Performance Battery; U&Es, creatinine and electrolytes.

for them (REC REF: 18/NE/0138). Participants will also be sought via Older Peoples Medicine Clinics, Day Units, assessment units and rehabilitation facilities in participating centres. Where available, existing information on handgrip strength and walk speed held in the clinical record will be used to identify potential participants. Finally, potential participants will be identified via screening of GP lists, in collaboration with the NIHR Primary Care Research Network. Where interrogation of the electronic FI (eFI) is possible,[51] invitations will be targeted to those with at least mild frailty denoted by an eFI of 0.12 or greater.

Potentially eligible participants will be sent or given a brief study information sheet, an invitation letter with a reply slip and a prepaid envelope to express interest. The reply slip will be returned to the central study team, who will pass on details to the site recruiting teams. Those who are interested will be contacted for prescreening as described below.

### Assignment of intervention

Participants will be randomised on a 1:1 basis using an interactive web-based randomisation system (Sealed Envelope). Minimisation (with a 30% random element) will ensure balance across the two arms based on the following stratification variables: sex and baseline walk speed ($\leq$0.6 or >0.6 m/s). The allocation sequence is prepared by Sealed Envelope and is concealed from the study team. Participants are allocated a pack number at randomisation; this pack number is used by the trial pharmacy to dispense prelabelled medication packs. Participants, the clinical team and the study team (including investigators, research nurses collecting outcomes data, senior statistician and trial manager) are blind to treatment allocation during the trial. For the independent data monitoring committee (IDMC), the statisticians preparing the report will be partially blind. This means that analysis will be conducted by arm, but the statisticians will not know the treatment allocation for each arm. They will become unblinded if the IDMC requests fully unblinded data. The randomisation system has functionality for emergency unblinding. This will only occur for valid medical or safety reasons where it is necessary for the treating clinician to know which treatment the participant has been receiving; the randomisation system is able to inform the treating clinician by email without unblinding other members of the study team or the participant.

### Data collection and management

The trial schedule of events is presented as a flow diagram (figure 2) and a table of trial processes (table 1). Recruited participants will be followed up for 4 months from the point of randomisation. Data including the number of participants identified, approached, prescreened and screened will be collected and documented on a site screening log. Data will be handled, computerised, stored and archived in accordance with the General Data Protection Regulation (2018), and the latest Directive

on GCP (2005/28/EC). Patient identifiable data will remain at each site and will not be collected as part of the trial dataset. Patient identification on data collection tools used will be through a unique sequential screening number allocated by site staff. Data will be transcribed by site staff from data collection worksheets to the trial's secure, password-limited, validated database (Sealed Envelope, London, UK). The participant trial record, including completed paper data collection tools, will be archived at site for 15 years following the end of the trial. Newcastle Clinical Trials Unit (NCTU) staff monitor trial conduct and data integrity; this is detailed in a Data Management Plan and a Monitoring Plan approved by the trial Sponsor.

### Statistical analysis

A statistical analysis plan will document the planned analyses and will be finalised prior to database lock at the end of the trial. For the primary outcome (4 m walk speed at 4 months postrandomisation), the difference between arms will be tested through fitting a linear regression model that is adjusted for baseline four metre walk speed and sex. A suitable transform of the data (eg, the best fitting Box-Cox transform) will be applied to ensure the data are approximately normally distributed. The model will allow estimation of the difference between arms together with the 95% CI and p value for testing the null hypothesis of no difference. Continuous secondary endpoints will be analysed in a similar manner to the primary outcome. Binary secondary endpoints will be analysed with a logistic regression model. All models will be adjusted for the outcome under test at baseline, baseline 4 m walk speed and sex.

The following preplanned subgroup analyses will be performed for the primary outcome: age >75 years vs$\leq$75 years, men versus women, baseline walk speed >0.6 m/s vs $\leq$0.6 m/s. In addition, a per-protocol analysis will be performed in participants with adherence to study medication $\geq$80%.

The primary analysis will be a complete-case analysis. In the event the mortality rate is >5%, we will perform a sensitivity analysis by imputing the worst score possible for the primary outcome (ie, 0 m/s) in participants who died before the 4-month visit. A further sensitivity analysis will be performed by imputing the primary outcome to the worst score possible in the event both mortality and withdrawal due to illness is >5%. If more than 10% of primary outcome data is missing after imputation for missing-ness due to death and withdrawal due to illness, a sensitivity analysis using multiple imputation via chained equations using the baseline covariates in the imputation model will be performed. Twenty imputed datasets will be used.

### Trial oversight

A trial management group (TMG), facilitated by NCTU, will convene approximately monthly throughout the duration of the trial. Members will consist of key NCTU staff, the Chief Investigator, coapplicants, trial statisticians, local

**Identification in Primary Care**

- GP records screened to identify potential participants
- Short trial summary, letter of invitation and reply slip sent to participants.

**Identification in Secondary Care**

- Potential participants identified from:
  - The SarcNet Registry
  - Older Peoples Medicine Clinics and Day Units
  - Newcastle BioResource
  - Rehabilitation Services
- Brief trial summary, letter of invitation and reply slip sent to participants.

**Pre-Screening** (telephone or video link):
Inclusion (SARC-F score, age), exclusion (diabetes diagnosis, current metformin prescription)

**Screening Visit**

- Consent interview with site Investigator (face to face, telephone or video link, facilitated by a Research Nurse)
- Screening assessments: demographics, medical history, concomitant medications, bloods (U&Es, LFTs), 4m walk speed, grip strength, 5x sit to stand, adverse events
- Eligibility confirmation

**Baseline Visit**

Height, weight, bloods (HbA1c, glucose, mechanistic outcomes), stool sample
Fried frailty score, Lean mass by bioimpedance, AGE skin auto fluorescence, SPPB, grip strength,  6 min walk
Quality of life questionnaires (Nottingham EADL, EQ5D-5L, SF-36), adverse events, concomitant medications

**Randomisation (n=80)**

**Placebo Arm (n=40)**

**Metformin Arm (n=40)**

**Placebo/Metformin packs issued/posted to participant**

**Phone call to participant** (1 week post-dispensing):
medication bottle cap check, date of first dose, adverse events, book 1 month safety visit

**Safety Visits** (1, 2 and 3 months  post-baseline visit):
bloods (U&Es, LFTs, glucose and lactate), adverse events, concomitant medications

**4 Month Follow Up Visit**
Concomitant medications, bloods (U&Es, LFTs, lactate, glucose, HbA1c, mechanistic outcomes), stool sample, adverse events, weight, Fried frailty index, Bio-impedance, AGE skin auto fluorescence, SPPB, grip strength, 6 min walk, quality of life questionnaires (Nottingham EADL, EQ5D-5L, SF-36 [1 week recall]), medication count, instructions for medication return.

**Figure 2**  MET-PREVENT trial flow diagram. EADL, extended activities of daily living; EQ5D-5L, EuroQoL 5 dimension 5 level; GPs, general practice; HbA1c, glycated haemoglobin; LFTs, liver function test; SF-36, Short Form 36; SPPB, Short Physical Performance Battery; U&Es, creatinine and electrolytes.

site research staff and a Sponsor representative. A patient and public involvement representative will attend TMG meetings at a frequency determined on an ongoing basis. The TMG undertakes, with the additional of external membership, the role of a Trial Steering Committee for this small, relatively low risk CTIMP study. An IDMC has been appointed to provide an independent review of participant safety. The independent members comprise two clinicians and a statistician. The IDMC will meet at least annually, and report directly to the TMG.

### Patient and public involvement

A patient panel (convened by VOICE Global) consisting of older people with lived experience of muscle weakness was involved in the design of the trial protocol, selection of outcome measures and reviewed the PIS. A lay member (SC) sits on the TMG and is involved in the overall management of the trial and in dissemination plans for the results.

### Harms

Data from all AEs will be recorded on the trial database at every trial visit. Serious AEs (SAEs) will be assessed for any relationship to the treatment intervention (causality), by a delegated, medically qualified site doctor. The following SAEs will be recorded on the database, but do not require reporting to Sponsor: any death or hospitalisation due to a new cardiovascular event, new diagnosis or treatment of cancer, fall or fracture, infection, delirium, reduced mobility, exacerbation of an existing medical condition, admission for elective or planned investigation or treatment, or hospitalisation due to nausea, vomiting, constipation or diarrhoea. The above exceptions to immediate SAE reporting refer only to SAEs where the trial medication is not deemed to be causally related to the event by the local site principal investigator. The chief investigator will assess, on behalf of sponsor, expectedness (by reference to the approved reference safety information) of any SARs.

Safety assessments will be conducted at screening and at the 1, 2, 3 and 4 months follow-up visits. Key safety data will include serum creatinine concentrations—eGFR will be calculated to ensure adequate renal function ($>30 \, mL/min/1.73 \, m^2$), LFTs, blood glucose and blood lactate concentrations. Lactate levels of $>4 \, mmol/L$ will lead to discontinuation of trial medication.

### Consent

Informed, written consent will be sought at the screening visit, prior to conducting any trial procedures, including screening assessments. Where the screening visit takes place in the participant's home, the informed consent discussion with a delegated local investigator may take place in person or remotely by telephone, teleconference or videoconference. A delegated research nurse will be present in the participant's home to facilitate this, and to countersign the consent form. The investigator will complete, date and sign a consent interview proforma to provide a comprehensive account of the remote consent interview.

### Ancillary and post-trial care

No provision for continuation of trial medication will be made by the trial team or sponsor. Metformin is not licensed for the indication under study in this trial; any off-licence use of metformin after the end of the trial would be the responsibility of the participant's usual primary or secondary care clinical team. Participants and their GP will be informed by letter of which treatment they took after all participants have completed their final visit and the database is locked.

## ETHICS AND DISSEMINATION
### Ethics approval

A favourable ethical opinion has been granted from the UK Health Research Authority Research Ethics Committee (North-West-Liverpool Central Research Ethics Committee; trial reference approval number 20/NW/0470). The trial has also received approval from the UK Medicines and Healthcare products Regulatory Agency (MHRA; trial reference number 2020-004023-16). The trial has been included in the National Institute for Health and care Research Clinical Research Network (NIHR CRN) portfolio (NIHR CRN study ID: 47772). The trial Sponsor is the Newcastle Upon Tyne Hospitals NHS Foundation Trust, Freeman Hospital, Freeman Road, High Heaton, Newcastle upon Tyne, NE7 7DN. The trial Sponsor has delegated responsibility for trial management to NCTU, including trial design; review and approval of all localised patient-facing documentation prior to implementation at each site; collection, analysis and interpretation of data; writing of the protocol publication and final clinical report manuscripts. This protocol manuscript is based on V.7.0 of the trial protocol (dated 13 December 2021)

### Dissemination policy

A final report of the trial will be provided to the Sponsor, Research Ethics Committee and the trial Funder within 1 year of the end of the study. The trial results will be uploaded to the European Union Drug Regulating Authorities Clinical Trials (EudraCT) database, as per the European Commission's guidelines on posting and publication of result-related information within 12 months. The trial is registered on the ISRCTN trial database and trial results will be made publicly available on the ISRCTN trial registry within 12 months of the end of the trial, defined as Last Patient Last Visit date.

The final clinical trial report will be used for publication and presentation at scientific meetings. Summaries of results will also be made available to investigators for dissemination within their clinical areas and to the wider public, and a summary of results will be sent to all participants along with their treatment allocation. If feasible within pandemic meeting restrictions, we will hold a

dissemination event for participants and their families to present and discuss the study results.

## Access to data

Access to the full blinded dataset will be limited to the TMG and to authors of the trial publication. At the end of the trial, a deidentified dataset will be prepared and stored by Newcastle University. Requests for data sharing with study teams outside Newcastle University or the study Sponsor, including international requests, will be considered by a data access committee with representation from the funder, sponsor and the trial chief investigator.

## DISCUSSION

Randomised controlled trials of treatments for sarcopenia and physical frailty are challenging to perform; previous trials have struggled to recruit their anticipated target numbers within the allocated time.[52] This is in part because sarcopenia (and to a lesser extent physical frailty) are not diagnoses commonly made or recorded in clinical practice and thus identifying people with these conditions is not straightforward. In addition, people with sarcopenia usually have multimorbidity[53] and by definition have components of the physical frailty syndrome. As a result, they may find it difficult to take part in clinical trials unless these trials are designed to facilitate their inclusion and retention.

We have sought to broaden the range of recruitment strategies employed to identify patients with sarcopenia and physical frailty. MET-PREVENT is one of the first sarcopenia trials to use a sarcopenia registry[31] to facilitate recruitment, and we also build on recent work deploying measures of muscle strength and walk speed into routine clinical practice to facilitate recruitment. Although recruitment through primary care lacks specificity for identifying those with sarcopenia or physical frailty, the use of the eFI can improve the specificity of searches, and large numbers of potentially eligible participants can be reached via this route. We have aimed to minimise the burden of study visits in MET-PREVENT, both in terms of visit duration and tasks to complete during each visit, but also by providing a flexible approach to the venue in which study visits are undertaken. Many patients with sarcopenia or physical frailty find it difficult to travel to study centres and some are unable to leave their home at all. Our flexible approach to study visits with many of them taking place in the homes of participants enables this group of patients who would normally be excluded from participation to take part in research.

Although the trial is powered to detect the minimum clinically important difference in four metre walk speed, it is not powered to detect clinically important differences for some of the secondary outcomes. However, the trial seeks to provide proof-of-concept for a larger, multicentre trial that would be powered to detect differences in these outcomes, including transitions from physical prefrailty to frailty. The trial population excludes some groups for safety reasons (eg, those with chronic kidney disease) which limits the generalisability of the results. Muscle biopsies are not being performed, thus direct information on changes in skeletal muscle structure and function will not be obtainable from this trial.

The collection of blood and stool samples for future mechanistic analyses will maximise the scientific value of the trial but also look to identify those groups that may be more likely to benefit from metformin as an intervention in a future large trial. In addition, the mechanistic analyses (which will be performed after the main trial is complete) will identify which mechanistic pathways are most important in mediating any benefit of metformin and in doing so will help us identify pathways and featured targets for other interventions to prevent or treat sarcopenia and physical frailty. We believe that this study design provides a template for other phase II intervention trials for sarcopenia and physical frailty; this template should both accelerate progress in translational research and also enable easier comparison of results across different trials.[54]

**Author affiliations**

[1]Newcastle Clinical Trials Unit, Newcastle University, Newcastle upon Tyne, UK
[2]NIHR Newcastle Biomedical Research Centre, Newcastle University and Newcastle Hospitals NHS Foundation Trust, Newcastle upon Tyne, UK
[3]Pharmacy Directorate, Newcastle Upon Tyne Hospitals NHS Trust, Newcastle Upon Tyne, UK
[4]Academic Unit of Elderly Care & Rehabilitation, University of Leeds, Bradford, UK
[5]Patient and Public Involvement Representative, Newcastle upon Tyne, UK
[6]Population Health Sciences Institute, Newcastle University, Newcastle upon Tyne, UK
[7]Gateshead Health NHS Foundation Trust, Gateshead, UK
[8]King's College London, London, UK
[9]Biosciences Institute, Newcastle University, Newcastle upon Tyne, UK

**Acknowledgements** The authors would like to thank the members of the MET-PREVENT Independent Data Monitoring Committee (DMC) for their continuing valuable expertise: (1) DMC Chair—Dr Terry Quinn (Senior Lecturer and Honorary Consultant, University of Glasgow) (2) DMC Statistician—Dr Lorna Aucott (Senior Statistician, Health Services Research Unit, University of Aberdeen) (3) DMC Clinician—Dr Victoria Haunton (Honorary Senior Lecturer/ Consultant Geriatrician, University of Plymouth/University Hospitals Plymouth NHS Trust).

**Contributors** MW is the trial chief investigator and senior author. MW, AAPS, AC, HCH, CM, CJS and TvZ led the funding application and protocol development. JW was the trial senior statistical advisor during the funding bid and advised on trial design. SH is the trial statistician. NW is the senior statistician and leads on the statistical analysis plan. PB is the sponsor pharmacy representative, and SC provided patient and public input. AJS, KJR, LM, LS and LR provided trial management and trial monitoring. KJR and MW drafted the manuscript for this publication. All authors contributed to protocol development and critical revision of the manuscript.

**Funding** This work was supported by the NIHR Newcastle Biomedical Research Centre, reference NU001533.

**Competing interests** None declared.

**Patient and public involvement** Patients and/or the public were involved in the design, or conduct, or reporting, or dissemination plans of this research. Refer to the Methods section for further details.

**Patient consent for publication** Not applicable.

**Provenance and peer review** Not commissioned; externally peer reviewed.

**ORCID iDs**
Miles Witham http://orcid.org/0000-0002-1967-0990
Andrew Clegg http://orcid.org/0000-0001-5972-1097
Helen C Hancock http://orcid.org/0000-0002-1494-8551
James Wason http://orcid.org/0000-0002-4691-126X
Nina Wilson http://orcid.org/0000-0001-5908-1720

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
