## [Reviewer comments · BMJ Open]

ARTICLE DETAILS

TITLE (PROVISIONAL)	MET-PREVENT: Metformin to improve physical performance in older people with sarcopenia and physical prefrailty/frailty – protocol for a double blind, randomised controlled proof of concept trial.
AUTHORS	Rennie, Katherine; Witham, Miles; Bradley, Penny; Clegg, Andrew; Connolly, Stephen; Hancock, Helen; Hiu, Shaun; Marsay, Leanne; McDonald, Claire; Robertson, Laura; Simms, Laura; Steel, Alison; Steves, Claire J.; Storey, Bryony; Wason, James; Wilson, Nina; von Zglinicki, Thomas; Sayer, Avan

VERSION 1 – REVIEW

REVIEWER	Lauretani, Fulvio Parma University Hospital, Internal Medicine and Subacute Unit
REVIEW RETURNED	15-Mar-2022

GENERAL COMMENTS	Dear Authors the protocol is well structured and presented
---

REVIEWER	Jung, Hee-Won Seoul National University Bundang Hospital, Geriatrics
REVIEW RETURNED	25-Mar-2022

GENERAL COMMENTS	I appreciate this opportunity to review a well-written protocol for intervention study using metformin for physical frailty. The aim and design of the study is clear to understand, and of interest for potential readers of the journal. I have only a single point that should be discussed before performing the trial. 1. Metformin is basically catabolic drug. Its pleiotropic actions including AMPK agonism, autophagy activation and mitochondrial hormesis all entails potentially catabolic consequences, and recent studies (both in rodents and human) suggest that metformin may prevent muscle protein synthesis. Hence, in human, metformin might be more beneficial to prevent adverse metabolic consequences and to extend lifespan via mechanism that is somehow similar to caloric restriction (research suggest there are some overlapping mechanisms and some non-overlapping mechanisms between metformin and CR) in rather young and over nourished persons. However, for already physically frail persons with decreased muscle performance, the basic premise of these anti-aging effect of metformin might be less likely beneficial, and further adversely effect on muscle function and mass. These physiological/biological heterogeneity due to human aging is one of underlying reason why higher protein consumption is recommended for frail, sarcopenic older adults, while high protein
--

	diet may accelerate aging speed in younger animals, as shown in studies using model organisms. The well-known TAME study also suffers this biological/clinical problem using metformin (a catabolic agent that might be helpful in extending lifespan in energy-abundant younger organisms) in older adults with aged phenotype (low physical performance, or age-related chronic conditions) who are suffering catabolic problems (anabolic resistance, higher GDF-15 concentration, net protein malnutrition etc). Hence, I suggest authors to expand and reinforce their mechanistic background using this drug for older adults with physical frailty/probable sarcopenia in revision.
--	--

VERSION 1 – AUTHOR RESPONSE

Reviewer 2 comment:

Metformin is basically catabolic drug. Its pleiotropic actions including AMPK agonism, autophagy activation and mitochondrial hormesis all entails potentially catabolic consequences, and recent studies (both in rodents and human) suggest that metformin may prevent muscle protein synthesis. Hence, in human, metformin might be more beneficial to prevent adverse metabolic consequences and to extend lifespan via mechanism that is somehow similar to caloric restriction (research suggest there are some overlapping mechanisms and some non-overlapping mechanisms between metformin and CR) in rather young and over nourished persons. However, for already physically frail persons with decreased muscle performance, the basic premise of these anti-aging effect of metformin might be less likely beneficial, and further adversely effect on muscle function and mass. These physiological/biological heterogeneity due to human aging is one of underlying reason why higher protein consumption is recommended for frail, sarcopenic older adults, while high protein diet may accelerate aging speed in younger animals, as shown in studies using model organisms. The well-known TAME study also suffers this biological/clinical problem using metformin (a catabolic agent that might be helpful in extending lifespan in energy-abundant younger organisms) in older adults with aged phenotype (low physical performance, or age-related chronic conditions) who are suffering catabolic problems (anabolic resistance, higher GDF-15 concentration, net protein malnutrition etc). Hence, I suggest authors to expand and reinforce their mechanistic background using this drug for older adults with physical frailty/probable sarcopenia in revision.

- Thank you for highlighting this line of argument. We agree that it is possible that metformin may have non-beneficial consequences on skeletal muscle and we have now added a paragraph and reference to discuss this and highlight the uncertainty that strengthens the need for a clinical trial in this area (p6 para 1)

We hope that the above changes satisfactorily address the points raised and we look forward to hearing your decision

VERSION 2 – REVIEW

REVIEWER	Jung, Hee-Won Seoul National University Bundang Hospital, Geriatrics
REVIEW RETURNED	16-Jun-2022
GENERAL COMMENTS	Thank you for the excellent revision that nicely addressing the point I provoked before.